# A Novel Antimicrobial Peptide Sp-LECin with Broad-Spectrum Antimicrobial Activity and Anti-*Pseudomonas aeruginosa* Infection in Zebrafish

**DOI:** 10.3390/ijms24010267

**Published:** 2022-12-23

**Authors:** Yan-Chao Chen, Wanlei Qiu, Weibin Zhang, Jingrong Zhang, Roushi Chen, Fangyi Chen, Ke-Jian Wang

**Affiliations:** 1State Key Laboratory of Marine Environmental Science, College of Ocean & Earth Sciences, Xiamen University, Xiamen 361102, China; 2State-Province Joint Engineering Laboratory of Marine Bioproducts and Technology, College of Ocean & Earth Sciences, Xiamen University, Xiamen 361102, China; 3Fujian Innovation Research Institute for Marine Biological Antimicrobial Peptide Industrial Technology, College of Ocean & Earth Sciences, Xiamen University, Xiamen 361102, China

**Keywords:** antimicrobial peptide, Sp-LECin, membrane permeability, *Pseudomonas aeruginosa*, antibacterial agent

## Abstract

New antimicrobial agents are urgently needed to address the increasing emergence and dissemination of multidrug-resistant bacteria. In the study, a chemically synthesized truncated peptide containing 22-amino acids derived from a C-type lectin homolog SpCTL6 of *Scylla paramamosain* was screened and found to exhibit broad-spectrum antimicrobial activity, indicating that it is an antimicrobial peptide (AMP), named Sp-LECin. Sp-LECin possessed the basic characteristics of most cationic AMPs, such as positive charge (+4) and a relatively high hydrophobicity (45%). After treatment with Sp-LECin, the disruption of microbial membrane integrity and even leakage of cellular contents was observed by scanning electron microscopy (SEM). In addition, Sp-LECin could bind lipopolysaccharide (LPS), increase the outer and inner membrane permeability and induce reactive oxygen species (ROS) production, ultimately leading to the death of *Pseudomonas aeruginosa*. Furthermore, Sp-LECin exhibited potent anti-biofilm activity against *P. aeruginosa* during both biofilm formation and maturation. Notably, Sp-LECin had no obvious cytotoxicity and could greatly improve the survival of *P. aeruginosa*-infected zebrafish, by approximately 40% over the control group after 72 h of treatment. This study indicated that Sp-LECin is a promising antibacterial agent with the potential to be used against devastating global pathogen infections such as *P. aeruginosa.*

## 1. Introduction

The discovery of antibiotics is considered one of the greatest achievements of 20th century medicine. Antibiotics were first introduced to the clinic as “wonder drugs” mainly because of their effectiveness in the treatment of serious bacterial infections. However, the global overuse and misuse of antibiotics in human therapy and livestock breeding over the past decades has given rise to the rapid dissemination of multidrug-resistant bacteria [1,2]. ESKAPE pathogens (*Enterococcus faecium*, *Staphylococcus aureus*, *Klebsiella pneumoniae*, *Acinetobacter baumannii*, *Pseudomonas aeruginosa*, and *Enterobacter* species) are responsible for the most nosocomial infections worldwide. Most of them are multidrug-resistant isolates, which pose a serious threat to human health. Polymyxin and vancomycin are usually used as the last-line of treatment for infections caused by gram-negative bacteria and gram-positive bacteria, respectively. Unfortunately, the emergence of polymyxin and vancomycin resistance in ESKAPE pathogens has become increasingly prevalent over the past few decades [3,4]. In 2017, the World Health Organization (WHO) published a global priority list of 12 antibiotic-resistant bacteria that are in urgent need of new antimicrobial agents [5]. Amongst these bacteria, ESKAPE pathogens were given the highest “priority” [5]. In this context, the development of alternatives to antibiotics, especially against ESKAPE pathogens, is particularly urgent. Through the unremitting efforts of researchers, a series of unconventional antimicrobial agents composed of peptides, nano-materials, synthetic polymers and probiotics have recently emerged to address the problem of antibiotic resistance [6,7,8,9].

Antimicrobial peptides (AMPs) are one of the important immune molecules of multicellular organisms. They are characterized as short peptides (typically within 100 amino acid residues in length) with activity against the invasion of exogenous pathogens, such as bacteria, fungi, viruses, and parasites [10]. In addition to direct antimicrobial activity, certain AMPs also display immunomodulatory properties [11]. AMPs are generally rich in cationic, hydrophilic, and hydrophobic amino acids. The cationic amino acid residues give AMPs a net positive charge, attracting them to negatively charged components on the bacterial surface, such as lipopolysaccharide (LPS), or lipoteichoic acid (LTA) via electrostatic interactions, followed by hydrophilic and hydrophobic amino acids form an amphipathic structure for their insertion into the bacterial membrane [12,13]. Most AMPs cause microbial death through non-specific interactions with membrane surfaces and disruption membrane integrity [13]. Besides, there is growing evidence that some AMPs can penetrate the cell membrane to reach their intracellular targets, where they inhibit intracellular processes including the biosynthesis of nucleic acids, protein and cell walls, protein-folding, proteases, cell division, etc. [14]. Due to these mechanisms of action, AMPs have different pharmacodynamics than traditional antibiotics and can eliminate microorganisms in a shorter time, thereby reducing the evolution of drug resistance in target microorganisms [15]. Interestingly, accumulating evidence shows that the synergistic effects between AMPs or between AMPs and antibiotics, and this combination can significantly reduce bacterial resistance, which may be a viable and effective strategy for the future treatment of ESKAPE pathogenic infections [16,17]. AMPs are considered attractive candidates for traditional antibiotics due to these favorable properties (such as broad-spectrum antimicrobial activities, multiple mechanisms of action, rapid killing effects, high capacity for synergies and low possibility of developing resistance), and some of them have already entered clinical trials [18].

To date, over 3400 AMPs derived from natural sources are listed in the Antimicrobial Peptide Database [19] since they were first characterized in the 1980s, and this number continues to grow. One of the classic ways to study AMPs is to extract and purify peptides from natural sources and then test their antimicrobial activity in vitro. In 2006, our group identified a novel AMP, scygonadin, which was isolated directly from the seminal plasma of the mud crab *Scylla paramamosain* using ion-exchange chromatography and reverse-phase liquid chromatography [20]. However, since it is usually difficult to purify large amounts of AMPs from natural sources, AMPs can be obtained by recombinant expression and chemical synthesis [21]. In 2009, we successfully obtained 65.9 mg/L of pure recombinant scygonadin by a prokaryotic expression system [22]. This recombinant scygonadin displayed antimicrobial activity against bacteria similar to that of the natural scygonadin, which allowed us to study the functions of scygonadin in depth [22]. Another strategy to study AMPs is to evaluate the antimicrobial activity of truncated fragments of a large protein. This strategy not only provides the key residues required for antimicrobial activity, but also reduces the cost of synthesizing the peptides. For example, Sph_12–38_, a truncated peptide of Sphistin from *S. paramamosain*, showed similar activity to Sphistin in most microorganisms [23]. Some AMPs undergoing clinical trials, such as OP-145 (phase I/II), P113 (phase II), LTX-109 (phase I/II), and EA-230 (phase II), have a lower number of amino acids as compared to their “original peptides” to reduce production costs [24].

C-type lectins (CTLs) are a superfamily of more than 1000 proteins defined with at least one CTL-like domain (CTLD) [25]. Most studies have focused on the functions of CTLs in innate and adaptive antimicrobial immune responses. In mammals, CTLs can be involved in a range of different physiological functions by recognizing self (endogenous) and non-self (exogenous) ligands, and can also regulate many essential processes as growth factors, opsonins, antimicrobial proteins, and components of the extracellular matrix [25]. In invertebrates, the immune function of CTLs is evolutionarily conserved and they participate in both cellular and humoral immunity through direct and indirect bactericidal effects [26,27,28]. In our previous study, we characterized a new CTL homolog SpCTL6 from *S. paramamosain* and found it could greatly improve the survival of *S. paramamosain* under bacterial infection [29]. In the present study, we designed a truncated peptide derived from the mature peptide of SpCTL6 and named it Sp-LECin. The antimicrobial activity of Sp-LECin was determined in vitro using a variety of bacteria (both gram-positive and gram-negative bacteria, including ESKAPE pathogens) and filamentous fungi. Furthermore, we investigated cell morphology, outer and inner membrane permeability, reactive oxygen species (ROS) generation, and LPS binding property to determine antimicrobial mechanism of Sp-LECin against *P. aeruginosa*. The anti-biofilm activity of Sp-LECin against *P. aeruginosa* was also determined. The cytotoxicity of Sp-LECin was performed to evaluate whether it could be safely used in in vivo experiments. Finally, the in vivo protective effect of Sp-LECin was investigated using zebrafish challenged with *P. aeruginosa*. This study aims to provide basic information for the development of an effective and biocompatible antibacterial agents.

## 2. Results

### 2.1. Antimicrobial Activity of Sp-LECin

Based on the analysis and prediction of the mature peptide of SpCTL6, a truncated peptide was identified and named as Sp-LECin (GCVFLLPAKPHNYKKVFLSKGV). Sp-LECin contains 22 amino acid residues, with a calculated molecular weight (MW) of 2.45 kDa, a total net charge of +4, an estimated isoelectric point (pI) of 9.87 and a hydrophobicity of 45% (Table 1), which is consistent with the basic characteristics of most cationic AMPs. Further experiments showed that Sp-LECin exerted a broad spectrum of antimicrobial activity and the minimum inhibitory concentration (MIC), minimum bactericidal concentration (MBC), and minimum fungicidal concentration (MFC) values measured are summarized in Table 2. Sp-LECin could inhibit the growth of a variety of gram-positive (*Listeria monocytogenes*, *E. faecium*, *Enterococcus faecalis*, *S. aureus*, *Staphylococcus epidermidis* and *Bacillus subtilis*) and gram-negative (*A. baumanii*, *P. aeruginosa*, *Pseudomonas stutzeri*, *Pseudomonas fluorescens*, *Escherichia coli* and *Shigella fiexneri*) bacteria, with MIC values in the range of 3–48 μM, and the MBC values lower than 48 μM. In addition, Sp-LECin could inhibit the conidial germination of several filamentous fungi (*Fusarium oxysporum*, *Fusarium solani*, *Fusarium graminearum*, *Aspergillus niger*, *Aspergillus fumigatus* and *Aspergillus ochraceus*) with MIC values in the range of 12–48 μM.

### 2.2. Morphological Changes in Microorganisms Treated with Sp-LECin

The scanning electron microscopy (SEM) analysis was employed to directly observe the morphological changes following with Sp-LECin treatment. The SEM images of the *E. faecium*, *A. baumannii*, *P. aeruginosa, P. stutzeri*, *A. niger* and *F. oxysporum* showed that Sp-LECin had a disruptive effect on microorganism surface (Figure 1). After treatment with 1 × MBC or 2 × MBC of Sp-LECin, the bacteria and filamentous fungi showed destruction of membrane integrity and even leakage of cellular contents. In contrast, the untreated cells exhibited a bright and smooth surface.

### 2.3. The Bactericidal Kinetics of Sp-LECin

Bactericidal kinetics assay was conducted to evaluate the bactericidal efficiency of Sp-LECin. As shown in Figure 2, Sp-LECin killed both gram-positive and gram-negative bacteria in a rapid concentration- and time-dependent manner. When incubated with *E. faecium*, Sp-LECin completely killed bacteria within 2 h at 1 × MBC and 2 × MBC (Figure 2A). Interestingly, Sp-LECin showed rapid killing activity against *A. baumannii*, *P. aeruginosa,* and *P. stutzeri,* and completely killed them within 45 min at 2 × MBC (Figure 2B–D). These results suggested that Sp-LECin had a rapid bactericidal effect.

### 2.4. Effect of Sp-LECin on the Membrane Permeability of P. aeruginosa and A. baumannii

Fluorescent probes N-phenyl-1-naphthylamine (NPN) and SYTOX Green were selected to determine the outer and inner membrane permeability in *P. aeruginosa* and *A. baumannii* after treatment with Sp-LECin. NPN is commonly used to evaluate the bacterial outer membrane permeability because its fluorescent intensity is weak in aqueous environments but strong in hydrophobic environments. SYTOX Green is an excellent green-fluorescent nucleic stain that penetrates only cells with damaged membranes. As shown in Figure 3A–D, the addition of Sp-LECin caused the fluorescence intensity of NPN and SYTOX Green to increase in a concentration- and time-dependent manner. Moreover, we conducted SYTO 9 and propidium iodide (PI) staining to determine inner membrane permeability in *P. aeruginosa* and *A. baumannii* after treatment with Sp-LECin (Figure 3E,F). When both SYTO 9 and PI are present, SYTO 9 can label all bacteria regardless of their membrane integrity, while PI can only label bacteria with damaged membranes, causing a reduction in SYTO 9 staining fluorescence. In this study, almost all cells in the untreated group showed green fluorescence and no red fluorescence, indicating that almost all bacterial cells were alive. Treatment with Sp-LECin resulted in an obvious increase in red fluorescence, suggesting a significant change in the inner membrane permeability. These results suggested that Sp-LECin could effectively enhance the outer and inner membrane permeability in *P. aeruginosa* and *A. baumannii.*

### 2.5. Binding Property of Sp-LECin to LPS

LPS, an abundant component of the outer membrane of gram-negative bacteria, is the target of many cationic AMPs. Therefore, we first investigated the effects of exogenous LPS on the antibacterial activity of Sp-LECin against *P. aeruginosa*. As shown in Figure 4A, the antibacterial activity of Sp-LECin was not affected when small amounts of exogenous LPS were added. However, an addition of 24 μg/mL of LPS inhibited the antibacterial activity of Sp-LECin. Furthermore, high levels of LPS (48 μg/mL) absolutely abolished the antibacterial activity of Sp-LECin. In addition, limulus ambocyte lysate (LAL) assay showed that Sp-LECin could inhibit LPS-induced activation of LAL enzyme in a dose-dependent manner, probably due to its ability to bind competitively to LPS (Figure 4B). This result revealed that Sp-LECin might exert its antibacterial effect by binding to LPS.

### 2.6. Effect of Sp-LECin on (Reactive Oxygen Species) ROS Production in P. aeruginosa

Excess ROS has multiple deleterious effects on different cell types, and induction of ROS production is also an important antimicrobial mechanism for many AMPs. 2,7-dichlorofuorescin diacetate (DCFH-DA) probe was used to determine ROS levels in *P. aeruginosa* upon Sp-LECin treatment. Intracellular ROS was monitored by incubating Sp-LECin at 12, 24 and 48 μM for 0.5 and 1 h, respectively. Exposure to Sp-LECin resulted in a significant increase in ROS levels in *P. aeruginosa*, which was positively correlated with time (Figure 5). These data suggested that Sp-LECin could induce the ROS accumulation in *P. aeruginosa* cells.

### 2.7. Anti-Biofilm Activity of Sp-LECin against P. aeruginosa

Because *P. aeruginosa* is able to form highly drug-tolerant biofilms, we investigated the activity of Sp-LECin against *P. aeruginosa*. The biofilm formation inhibition upon Sp-LECin treatment was quantified using crystal violet (CV) staining and the results were shown in Figure 6A. The concentration of Sp-LECin required to inhibit *P. aeruginosa* biofilm formation was 48 μM and the inhibition rate was more than 90%. In addition, the cell viability of the preformed *P. aeruginosa* biofilms treated with Sp-LECin was measured by the resazurin assay. As shown in Figure 6B, respiration of *P. aeruginosa* in preformed biofilms could be significantly inhibited by 25.5% and 71.9% in 24 μM and 48 μM of Sp-LECin, respectively. These results suggested that Sp-LECin exhibited potent anti-biofilm activity against *P. aeruginosa*.

### 2.8. Efficacy of Sp-LECin Treatment on P. aeruginosa Infection in a Zebrafish Mode

The cytotoxicity of Sp-LECin was analyzed using different cell lines to evaluate whether Sp-LECin could be safely used for in vivo experiments. As shown in Figure 7A–E, Sp-LECin showed no cytotoxicity to crab hemocytes and mammalian cells, and no hemolytic activity to mouse red blood cells, indicating that Sp-LECin had good biocompatibility. To assess the effectiveness of Sp-LECin in vivo, zebrafish were intraperitoneally infected with *P. aeruginosa* (approximately 3 × 10^5^ CFU/fish) and treated with Sp-LECin. As shown in Figure 7D, the cumulative mortality rates at 72 h post treatment were 30% and 70% in the Sp-LECin-treated and control groups, respectively. After statistical analysis, the survival rate of zebrafish in the Sp-LECin treatment group was significantly higher than that of the control group (*p* = 0.0064). These results indicated that Sp-LECin was effective in controlling *P. aeruginosa* infection in zebrafish.

## 3. Discussion

AMPs are one of the important immune molecules of innate immunity. They are small proteins with antibacterial, antifungal, antiviral, and immunoregulatory activities, which form the first line of defense against invasion by exogenous pathogen. Due to their broad spectrum of activities, multiple mechanisms of action, rapid bactericidal effect, high synergistic capacity and low possibility for resistance development, AMPs are considered attractive candidates for conventional antibiotics, and several have already entered clinical trials [18]. In our previous study, we characterized a new C-type lectin homolog SpCTL6 from the mud crab *S. paramamosain* [29]. Although SpCTL6 had no obvious antimicrobial activity in vitro, it could greatly improve the survival of *S. paramamosain* under bacterial infection [29]. In the present study, we designed a truncated peptide derived from the mature peptide of SpCTL6 and named Sp-LECin. We found that micromolar levels of Sp-LECin displayed potent antimicrobial activity against bacteria and filamentous fungi. Sp-LECin rapidly killed *P. aeruginosa* through increasing the permeability of outer and inner membranes and inducing intracellular ROS accumulation. Sp-LECin also showed good anti-biofilm activity against *P. aeruginosa*. Additionally, Sp-LECin did not exhibit cytotoxicity and had a significant anti-infective effect in zebrafish. Therefore, Sp-LECin is a potential candidate for antimicrobial drug development.

*P. aeruginosa* is a common, gram-negative opportunistic pathogen. It is usually the most common cause of respiratory and urinary tract infections, blood infections, burn infections, and external ear infections. Treatment of *P. aeruginosa* infections has become increasingly difficult due to its remarkable resistance to antibiotics. *P. aeruginosa* can perform multiple mechanisms to counter most antibiotics, including reducing antibiotic uptake, altering antibiotic targets, overexpressing efflux pumps, and forming biofilms and persistent cells, etc. [30]. The increasing prevalence of infections caused by multidrug-resistant or extensively drug-resistant strains of *P. aeruginosa* is associated with severe morbidity and mortality [31]. New antimicrobial agents are urgently needed to address the growing threat of *P. aeruginosa*. Nowadays, many AMPs have been tested against *P. aeruginosa* and different modes of action have been described. For instance, CM4 mainly destroys the cell membrane [32]; DP7 interacts with LPS of the outer membrane [33]; AS-hepc3_(48–56)_ alters the permeability of the outer and inner membranes [34]; lipopeptide polymyxin B interacts with LPS, disrupts cellular osmotic balance and induces endogenous ROS production [35]. In the present study, we found that Sp-LECin displayed potent growth inhibition and rapid bactericidal activity against *P. aeruginosa* at low concentrations (MIC, 12–24 μM; MBC, 12–24 μM). The SEM observation showed that Sp-LECin disrupted cellular integrity and even caused leakage of *P. aeruginosa* cell contents. The NPN uptake assay and SYTOX Green uptake assay further confirmed that Sp-LECin could penetrate the outer and inner membrane of *P. aeruginosa*. Additionally, exogenous LPS could reduce the antibacterial effect of Sp-LECin against *P. aeruginosa*. It is speculated that the exogenous LPS binds to Sp-LECin, thereby reducing the aggregation of Sp-LECin in *P. aeruginosa*. These results clearly demonstrated that Sp-LECin can target and disrupt the membranes, which is similar to other AMPs, such as DP7 [33] and AS-hepc3_(48–56)_ [34]. This mechanism of bactericidal action might be associated with the physicochemical properties of Sp-LECin, a cationic peptide with a total net charge of +4 and a total hydrophobicity of 45%, allowing it to interact with negatively charged components of the microbial cell surface, such as LPS and disrupt membrane integrity. These results indicated that Sp-LECin is expected to be a promising antimicrobial drug for future use in the control *P. aeruginosa* infection.

ROS, such as hydrogen peroxide (H_2_O_2_), hydroxyl radicals (OH·) and superoxide anions (O_2_^−^) are commonly produced in cellular metabolism and they are essential for homeostasis and cellular signaling [36]. In fact, ROS function as “double-edged swords”. Excess ROS can induce oxidative stress and cause deleterious oxidative damage to DNA, proteins, and cell membranes [37]. Several studies have shown that AMPs exert their antimicrobial activity through the production of ROS. When bacteria are exposed to antimicrobial agents, lethal antimicrobial stress can trigger the accumulation of intracellular ROS, causing further damage to bacteria [38]. Lactoferricin B like peptide (LBLP), a cationic peptide derived from *Scolopendra subspinipes mutilans*, exerted antibacterial activity against *E. coli* by producing ROS leading to apoptosis-like cell death [39]. HD5d5, a simplified derivative of human defensin, showed its antibacterial activity against *A. baumannii* by reducing the activities of superoxide dismutase and catalase, and ultimately increasing the microbially unfavorable ROS levels [40]. In the present study, we speculated that the cell membrane is the primary target of Sp-LECin, which first binds to LPS, interacts with the bacterial cell membrane, and disrupts its integrity. Subsequently, the damaged cell membrane induces the production of intracellular ROS, causing further damage to *P. aeruginosa.* This ROS-associated antibacterial mechanism is worthy of further investigation.

Biofilms are aggregates of microorganisms that attached to solid surfaces and enclosed in an extracellular polymeric substance matrix, including exopolysaccharides, proteins, metabolites and extracellular DNA [41]. Bacterial cells within biofilm can escape the host immune system and are 100–1000 times more resistant to antibiotics than their planktonic counterparts [42,43]. *P. aeruginosa* cells are well known for its ability to attach to the surface of indwelling medical devices, such as urinary catheters, implants, and contact lenses, to form biofilms [44]. They can develop high-levels of biofilm-specific resistance to different classes of antibiotics, such as tobramycin, gentamicin, ciprofloxacin, nalidixic acid, chloramphenicol, ampicillin, piperacillin, and kanamycin, which greatly reduces treatment options [45]. Common mechanisms of biofilm-mediated resistance include slowing the growth of biofilm cells, inducting adaptive stress responses, and forming multidrug-tolerant persistent cells [30]. Interestingly, Sp-LECin showed good anti-biofilm activity against *P. aeruginosa*, which not only effectively suppressed the formation of biofilm, but also inhibited the growth of preformed biofilms. These results suggested that Sp-LECin may be able to address the clinical challenges posed by *P. aeruginosa* biofilms.

Although AMPs are one of the most promising alternatives to conventional antibiotics, there are still some obstacles that hinder their therapeutic use. For example, most AMPs, especially α-helical AMPs that act on membranes, are not fully selective for microbial cells and some AMPs are toxic to mammalian cells [46,47]. Furthermore, under physiological conditions in vivo (such as environmental pH, presence of salts, serum and proteases, etc.), some AMPs may lose their antimicrobial activity [48,49,50]. Therefore, we first analyzed the cytotoxicity and hemolytic activity of Sp-LECin before evaluating its antimicrobial activity in vivo. It was found that Sp-LECin had no toxicity toward crab hemocytes and mammalian cells and no hemolytic activity toward mouse red blood cells, indicating that it could be safely used for in vivo experiments. As reported, the optimum net charge and hydrophobicity in the AMPs characterized to date is +4 to +6 and 40% to 60%, respectively, beyond which AMPs not only do not increase antimicrobial activity but also lead to an increase in toxicity [51]. In the present study, Sp-LECin contains 22 amino acid residues, including 4 Lys residues, with a total net charge of +4 and a total hydrophobicity of 45%, which is consistent with the basic characteristics of most cationic AMPs. We hypothesized that the appropriate net charge and hydrophobicity made Sp-LECin have no obvious cytotoxicity. In addition, regular and stable secondary structures usually lead to increased hemolytic activity and cytotoxicity of AMPs. For example, melittin and LL-37 exhibit strong antimicrobial activity but high cytotoxicity or hemolytic activity, probably due to their stable α-helical structure [52,53]. Whether the secondary structure of Sp-LECin is specifically related to its low toxicity needs further study. Then, we established a *P. aeruginosa* infection model in zebrafish to test the antimicrobial effect of Sp-LECin in vivo. The survival curve of zebrafish showed that the experimental group receiving Sp-LECin injection had a 70% survival rate after 72 h of treatment, which was significantly higher than that of the control group (30%). These results indicated that Sp-LECin is safe and effective in controlling *P. aeruginosa* infection in zebrafish, suggesting that Sp-LECin is promising as an anti-infective drug.

## 4. Materials and Methods

### 4.1. Strains and Culture Conditions

All commercially available strains used in this study were purchased from China General Microbiological Culture Collection Center (CGMCC), including *L. monocytogenes* (CGMCC NO. 1.10753), *E. faecium* (CGMCC NO. 1.131), *E. faecalis* (CGMCC NO. 1.2135), *S. aureus* (CGMCC NO. 1.879), *S. epidermidis* (CGMCC NO. 1.4260), *B. subtilis* (CGMCC NO. 1.3358), *A. baumanii* (CGMCC NO. 1.6769), *P. aeruginosa* (CGMCC NO. 1.2421), *P. stutzeri* (CGMCC NO. 1.1803), *P. fluorescens* (CGMCC NO. 1.3202), *E. coli* (CGMCC NO. 1.2389), *S. fiexneri* (CGMCC NO. 1.1868), *F. oxysporum* (CGMCC NO. 3.6785), *F. solani* (CGMCC NO. 3.5840), *F. graminearum* (CGMCC NO. 3.3490), *A. niger* (CGMCC NO. 3.316), *A. fumigatus* (CGMCC NO. 3.5835) and *A. ochraceus* (CGMCC NO. 3.5830). Bacterial strains grew at 37 °C in nutrient broth (OXOID, Hampshire, UK) and fungal strains were cultured in potato dextrose agar (Hope Bio, Qingdao, China) at 28 °C, respectively.

### 4.2. Truncated Peptide Design and Chemical Synthesis

The mature peptide of SpCTL6 gene (a C-type lectin gene from mud crab *S. paramamosain*, Genbank NO. MW119266) was subjected to the antimicrobial peptide database (Collection of Anti-Microbial Peptides, CAMP_R3_) [54] to screen for fragments with high probability of AMP. A truncated 22-amino acid peptide (GCVFLLPAKPHNYKKVFLSKGV) from residue Gly^107^ to residue Val^128^ of SpCTL6 mature peptide was selected as a candidate for AMP, and named Sp-LECin. The physicochemical parameters of Sp-LECin, including MW, net charge, pI and hydrophobicity were calculated by ProtParam [55] and Antimicrobial Peptide Database [19]. Sp-LECin was chemically synthesized by Genscript (Nanjing, China) with a purity of >95% and verified by HPLC and mass spectrometry (Appendix A).

### 4.3. Antimicrobial Activity Assay

The antimicrobial activity assay was performed by broth microdilution and agar plating methods as previously reported [34,56]. Briefly, logarithmic growth-phase cultures of bacteria were harvested and diluted to approximately 2 × 10^6^ CFU/mL in 10 mM sodium phosphate buffer (NaPB, pH 7.4), supplemented with 40% Mueller-Hilton broth (MHB), and the conidia of filamentous fungi were harvested and diluted to approximately 2 × 10^4^ conidia/mL in 10 mM NaPB supplemented with 40% potato dextrose water (PDW). Subsequently, the microbial suspensions were aliquoted into 96-well microplate and incubated with serial dilutions of Sp-LECin (from 3 to 96 μM). The MIC, MBC, and MFC values were determined as previously described [56]. This assay was carried out in triplicate at least three times.

### 4.4. Scanning Electron Microscope Assay

The effects of Sp-LECin on *E. faecium, A. baumannii, P. aeruginosa, P. stutzeri, A. niger,* and *F. oxysporum* were observed using SEM as previously described [57]. Briefly, the bacteria and conidia of filamentous fungi were harvested and diluted to approximately 1 × 10^8^ CFU/mL in 10 mM NaPB supplemented with 40% MHB and 2 × 10^5^ cells/mL in 10 mM NaPB supplemented with 40% PDW, respectively, and incubated with 1 × MBC and 2 × MBC of Sp-LECin for 1 h. After incubation, the cells were fixed with 2.5% (vol/vol) glutaraldehyde, washed three times with PBS, and then placed on poly-L-lysine coated glass slides. Subsequently, the samples were dehydrated in a series of graded ethanol and critical point dried using the automated dryer (Leica EM CPD300, Wetzlar, Germany). Finally, the samples were coated with gold and observed under a field emission scanning electron microscope (Zeiss SUPRA 55, Oberkochen, Germany).

### 4.5. Bactericidal Kinetics Curve

A gram-positive bacteria (*E. faecium*) and three gram-negative bacteria (*A. baumannii*, *P. aeruginosa* and *P. stutzeri*) were selected to determine the bactericidal kinetics of Sp-LECin following prior description [58]. Briefly, bacterial suspensions were prepared as described for the antimicrobial assay and incubated with 1 × MBC and 2 × MBC of Sp-LECin at 37 °C. At different time points after co-incubation, the cultures were sampled, diluted, and spread on nutrient agar. The sterilization index was calculated using the following formula: sterilization index (%) = recovered CFU/initial CFU × 100%, where initial CFU refers to the number of colonies at 0 min and recovered CFU refers to the number of colonies at different sampling points.

### 4.6. Outer Membrane Permeability Assay

The ability of Sp-LECin to permeate the outer membrane of *P. aeruginosa* and *A. baumannii* was measured by the NPN (Sigma, St. Louis, MO, USA) uptake assay as previously described with some modifications [59]. Briefly, the bacterial cells in the logarithmic phase were harvested, washed, and then suspended in 5 mM HEPES buffer to a final cell density of approximately 1 × 10^8^ CFU/mL. The fluorescent probe NPN dissolved in 95% ethanol (final concentration of 10 μM) was added to the suspension. Subsequently, the above bacterial suspension was aliquoted into a 96-well microplate and incubated with different concentrations of Sp-LECin (from 6 to 48 μM). LL-37 (GL Biochem, Shanghai, China) and polymyxin B (Solarbio, Beijing, China) were used as positive controls for permeabilizing outer membrane of *P. aeruginosa* and *A. baumannii*, respectively. Finally, fluorescence was recorded every 2.5 min in a microplate reader (Tecan, Männedorf, Switzerland) with excitation and emission wavelengths of 350 and 420 nm, respectively.

### 4.7. Inner Membrane Permeability Assay

The penetration of Sp-LECin into the inner membrane of *P. aeruginosa* and *A. baumannii* was measured by SYTOX Green (Invitrogen, Carlsbad, CA, USA) uptake assay according to a previous report with the following modifications [60]. Briefly, the bacterial cells diluted to approximately 1 × 10^8^ CFU/mL in 10 mM NaPB supplemented with 40% MHB were prepared as described for SEM assay and then the fluorescent dye SYTOX Green was added to the suspension at a final concentration of 5 μM. Different concentrations of Sp-LECin (from 6 to 48 μM) were incubated with bacterial cells in a 96-well microplate for analysis. LL-37 and polymyxin B were used as positive controls for permeabilizing inner membrane of *P. aeruginosa* and *A. baumannii*, respectively. Fluorescence intensity was recorded every 2.5 min in a microplate reader (Tecan, Männedorf, Switzerland) with excitation and emission wavelengths of 485 and 530 nm, respectively.

### 4.8. Live-Dead Staining Assay

Dead bacteria caused by Sp-LECin were evaluated by live-dead staining according to the protocol of the LIVE/DEAD BacLight Bacterial Viability Kit (Invitrogen, Carlsbad, CA, USA). Briefly, *P. aeruginosa* and *A. baumannii* cells diluted to approximately 1 × 10^8^ CFU/mL in 10 mM NaPB supplemented with 40% MHB were prepared as described for SEM assay and incubated with Sp-LECin at concentration of 24 μM and 48 μM. LL-37 and polymyxin B served as positive control. After incubation, the cells were washed twice with PBS and stained with SYTO 9 and PI for 15 min in the dark. Fluorescent mages were obtained with a confocal laser scanning microscopy (Zeiss, Oberkochen, Germany).

### 4.9. LPS Binding Assay

The effect of LPS on the antimicrobial activity of Sp-LECin was evaluated using broth microdilution assay as previously reported with slight modifications [61]. Briefly, LPS (final concentration of 0–48 μg/mL, 25 μL) from *P. aeruginosa* (Sigma, St. Louis, MO, USA) and Sp-LECin (final concentration of 24 μM, 25 μL) were added into a 96-well microplate. Subsequently, 50 μL of *P. aeruginosa* suspension (approximately 2 × 10^6^ CFU/mL) prepared as described for antimicrobial assay was added to each well. The absorbances at the wavelength of 600 nm were recorded every half hour at 37 °C in a microplate reader (Tecan, Männedorf, Switzerland). In addition, the ability of Sp-LECin to bind LPS was assessed using a ToxinSensor™ Chromogenic LAL Endotoxin Assay Kit (Genscript, Nanjing, China) as previously described [62]. Briefly, 50 μL of Sp-LECin was incubated with 50 μL of endotoxin (1 endotoxin unit) in endotoxin-free vials for 30 min at 37 °C. Afterwards, 100 μL of LAL reagent was added to each vial, and the mixture was incubated at 37 °C for 9 min. Then, 100 μL of LAL chromogenic substrate was added to each vial. After incubation at 37 °C for 6 min, the stop solution and color-stabilizer solution were added. Finally, 200 μL of the final solution was transferred into a 96-well microplate to read the absorbance at 545 nm.

### 4.10. Intracellular ROS Measurement

The levels of ROS in Sp-LECin-treated *P. aeruginosa* were quantified using the fluorescent probe DCFH-DA (Nanjing Jiancheng Bioengineering Institute, Nanjing, China), as previously described [63]. *P. aeruginosa* cells diluted to approximately 1 × 10^8^ CFU/mL in 10 mM NaPB supplemented with 40% MHB were prepared as described for SEM assay and incubated with different concentrations of Sp-LECin (from 12 to 48 μM) or 12 μM of LL-37. After incubation, the cells were washed with PBS and then stained with 10 μM of DCFH-DA for 30 min at 37 °C. Finally, fluorescence intensity was measured with excitation and emission wavelengths of 488 and 525 nm, respectively, using a microplate reader (Tecan, Männedorf, Switzerland).

### 4.11. Anti-Biofilm Assay

The anti-biofilm assays were performed in 96-well microplates according to a previous report with some modifications [64]. For the biofilm formation assay, suspensions of *P. aeruginosa* diluted to approximately 1 × 10^8^ CFU/mL in 10 mM NaPB supplemented with 40% MHB were prepared as described for SEM assay and incubated with different concentrations of Sp-LECin (from 6 to 48 μM) in a 96-well microplate. The microplate was incubated at 37 °C for 24 h without shaking to allow biofilm formation. The biofilm mass was measured by CV staining as previously described [64]. In experiments with preformed biofilms, suspensions of *P. aeruginosa* diluted to approximately 1 × 10^6^ CFU/mL in 10 mM NaPB supplemented with 40% MHB were aliquoted into 96-well microplate and incubated at 37 °C for 24 h without shaking to allow biofilm formation. After washed with PBS, the MHB medium containing 0.1 mM of resazurin was added to the wells and then treated with serial dilutions of Sp-LECin (0–48 μM). The microplates were incubated at 37 °C for 6 h. The respiratory activity of cells in biofilms was evaluated by a modified resazurin assay as previously described [64].

### 4.12. Cytotoxicity and Hemolytic Activity

The cytotoxicity of Sp-LECin was determined on crab hemocytes, human embryonic kidney cells (HEK-293T), human hepatic cells (L02), and mouse embryonic fibroblast (3T3). Briefly, 100 μL of crab hemocytes, HEK-293T, or L02 cells were seeded at 10^4^ cells/well in 96-well microplates and incubated overnight at 37 °C with 5% CO_2_ (crab hemocytes, 26 °C without 5% CO_2_). Then, the cells were incubated in a culture medium supplemented with various concentrations of Sp-LECin (0, 3, 6, 12, 24, and 48 μM). After incubation at 26 °C or 37 °C for 24 h, cell viability was assessed using MTS-PMS assay. The hemolytic activity of Sp-LECin was determined using fresh mouse red blood cells. All animal procedures were carried out in strict accordance with the National Institute of Health Guidelines for the Care and Use of Laboratory Animals and were approved by the Animal Welfare and Ethics Committee of Xiamen University. Briefly, mouse red blood cells were harvested, washed three times with 0.9% saline, and resuspended in the same saline to a concentration of 4% (*v/v*). Then, 100 μL of mouse red blood cells and 100 μL of Sp-LECin solution were added into 96-well microplate. After incubated at 37 °C for 1 h without shaking, samples were centrifuged at 4000 rpm for 3 min. The supernatant from each well was transferred to a new 96-well microplate and the absorbance was measured at 540 nm. 1% Triton X-100 was served as a positive control. The percentage of hemolysis was calculated using the following formula: Haemolysis (100%) = (A_540_Sample–A_540_Blank)/(A_540_Positive controls–A_540_Blank) × 100.

### 4.13. Evaluation of the In Vivo Activity of Sp-LECin on Zebrafish Infected with P. aeruginosa

The in vivo antibacterial activity of Sp-LECin was evaluated using a zebrafish infection model. The animal experiment was carried out in strict accordance with the guidelines of Xiamen University. First, zebrafish were anesthetized with 100 µg/mL tricaine methanesulfonate (MS222), and then 8.4 μL of *P. aeruginosa* (approximately 3 × 10^5^ CFU/mL) was injected intraperitoneally into the zebrafish. The zebrafish were randomly divided into two groups (control group and treatment group) with 20 fish in each group. Half an hours after infection, fish in the treatment group was injected with 8.4 µL of 1.5 mg/mL Sp-LECin, while fish in the control group were injected with an equal volume of PBS. The cumulative mortality of the fish was observed and recorded at different time point over a period of 72 h.

### 4.14. Statistical Analysis

All experimental data were analyzed using GraphPad Prism 6 (GraphPad Software Inc., San Diego, CA, USA) and SPSS 25 (IBM Corp., Armonk, NY, USA). Two-tailed *p*-values were used for all analyses, and a *p*-value < 0.05 was considered statistically significant. All data were represented as mean ± standard error of mean.

## 5. Conclusions

In summary, based on the analysis and prediction of mature peptide of SpCTL6, a truncated peptide was identified and named Sp-LECin. This peptide showed a broad-spectrum antimicrobial activity against a variety of gram-positive bacteria, gram-negative bacteria and filamentous fungi. It was found that Sp-LECin can effectively kill *P. aeruginosa* via multiple mechanism of bactericidal action including increasing outer and inner membrane permeability, and inducing ROS accumulation. Sp-LECin had good biocompatibility, and was not toxic to crab hemocytes and mammalian cells. These data suggest that the AMP Sp-LECin could be useful for the future development of therapeutic agents effective against bacterial infections.

## Figures and Tables

**Figure 1 ijms-24-00267-f001:**
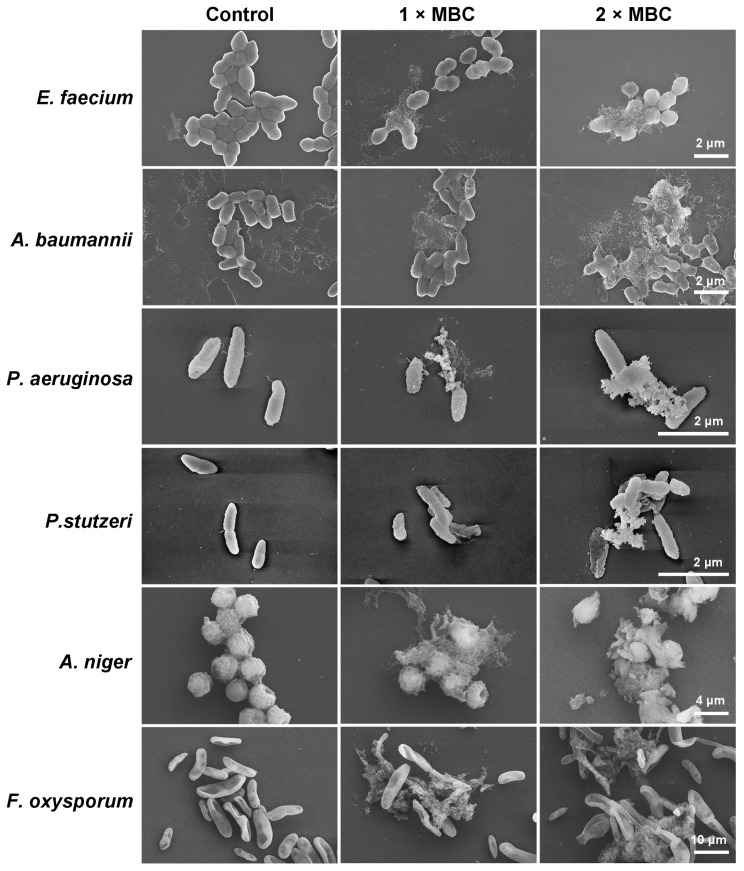
Effect of Sp-LECin on morphological and structural changes of *E. faecium*, *A. baumannii*, *P. aeruginosa*, *P. stutzer, A. niger* and *F. oxysporum* observed by a scanning electron microscopy (SEM).

**Figure 2 ijms-24-00267-f002:**
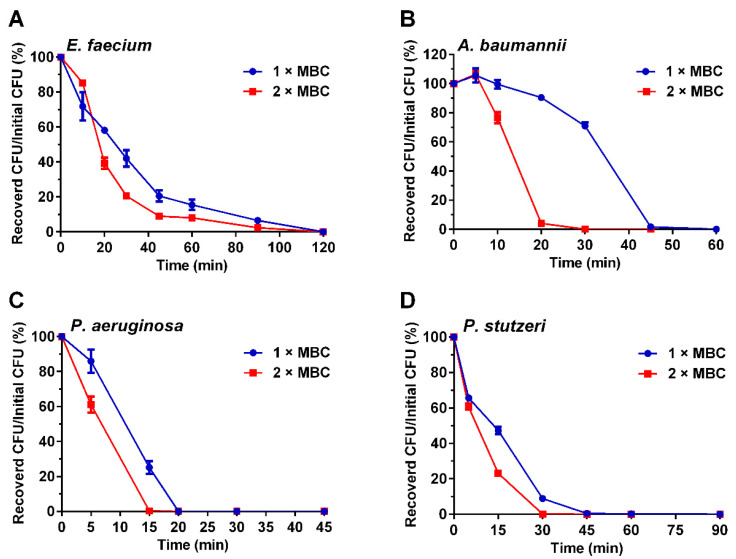
The bactericidal kinetics of Sp-LECin against *E. faecium* (**A**), *A. baumannii* (**B**), *P. aeruginosa* (**C**) and *P. stutzeri* (**D**) at 1×, and 2× MBC. Data represent mean ± standard error of mean from three independent biological replicates.

**Figure 3 ijms-24-00267-f003:**
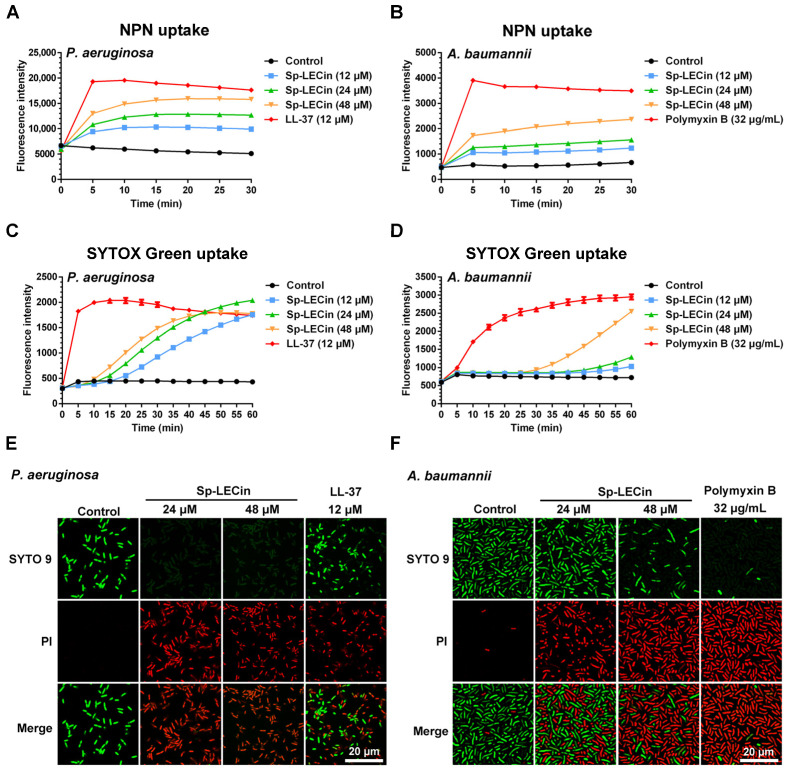
Effect of Sp-LECin on the membrane permeability of *P. aeruginosa* and *A. baumannii*. (**A**,**B**) Outer membrane permeability after Sp-LECin treatment was measured by N-phenyl-1-naphthylamine (NPN) uptake assay. The NPN fluorescence was recorded at the excitation and emission wavelengths of 350 and 420 nm, respectively. (**C**,**D**) Inner membrane permeability after Sp-LECin treatment was measured by SYTOX Green uptake assay. The SYTOX Green fluorescence was recorded at the excitation and emission wavelengths of 485 and 530 nm, respectively. The bars indicate the mean ± standard error of mean (*n* = 3). (**E**,**F**) Confocal laser-scanning microscope (CLSM) images representing the cell membrane permeability by SYTO 9 and propidium iodide (PI) staining in *P. aeruginosa* and *A. baumannii* cells treated with Sp-LECin.

**Figure 4 ijms-24-00267-f004:**
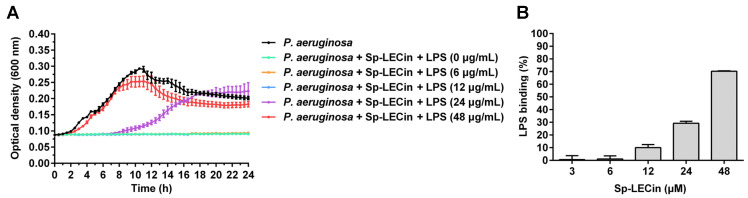
The binding property of Sp-LECin to LPS. (**A**) The Effect of exogenous LPS (0, 6, 12, 24 and 48 μg/mL) on the antibacterial activity of Sp-LECin (24 μM) was assessed by measuring the growth curve of *P. aeruginosa*. The bars indicate the mean ± standard error of mean (*n* = 3). (**B**) Determination of the binding property of Sp-LECin to LPS by the LAL assay. The bars indicate the mean ± standard error of mean (*n* = 5).

**Figure 5 ijms-24-00267-f005:**
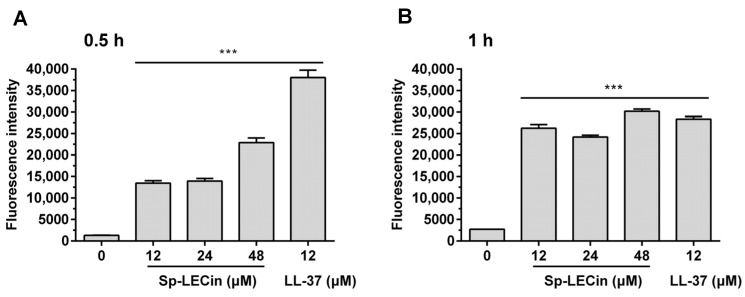
Effect of Sp-LECin on ROS production in *P. aeruginosa*. Intracellular ROS was monitored using 2,7-dichlorofuorescin diacetate (DCFH-DA) probe after treatment with different concentrations of Sp-LECin (0, 12, 24 and 48 μM) for 0.5 (**A**) and 1 h (**B**), respectively. LL-37 was used as a positive control. Significant difference between control group and AMP treatment group was indicated by asterisks as *** *p* < 0.001. The bars indicate the mean ± standard error of mean (*n* = 5).

**Figure 6 ijms-24-00267-f006:**
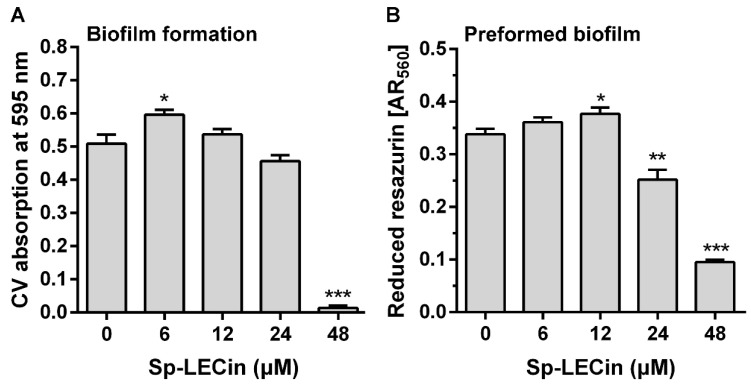
Anti-biofilm activity of Sp-LECin against *P. aeruginosa*. (**A**) The inhibitory effect of Sp-LECin on the formation of *P. aeruginosa* biofilm. The biofilm mass was quantified by crystal violet (CV) and the absorbance at 595 nm was measured. (**B**) The inhibitory effect of Sp-LECin against the preformed biofilm of *P. aeruginosa*. The amount of reduced resazurin and the residual amount of oxidized resazurin was determined by measuring at the absorbance at 560 nm and 620 nm, respectively. The corrected A560 value (AR560) was calculated using the following formula: AR560 = A560 − (A620 × RO) and RO = AO560/AO620, where A560 and A620 are sample absorbance and AO560 and AO620 are the absorbance of medium containing 0.1 mM resazurin. Significant difference between control group and Sp-LECin treatment group was indicated by asterisks as * *p* < 0.05, ** *p* < 0.01 and *** *p* < 0.001. The bars indicate the mean ± standard error of mean (*n* = 5).

**Figure 7 ijms-24-00267-f007:**
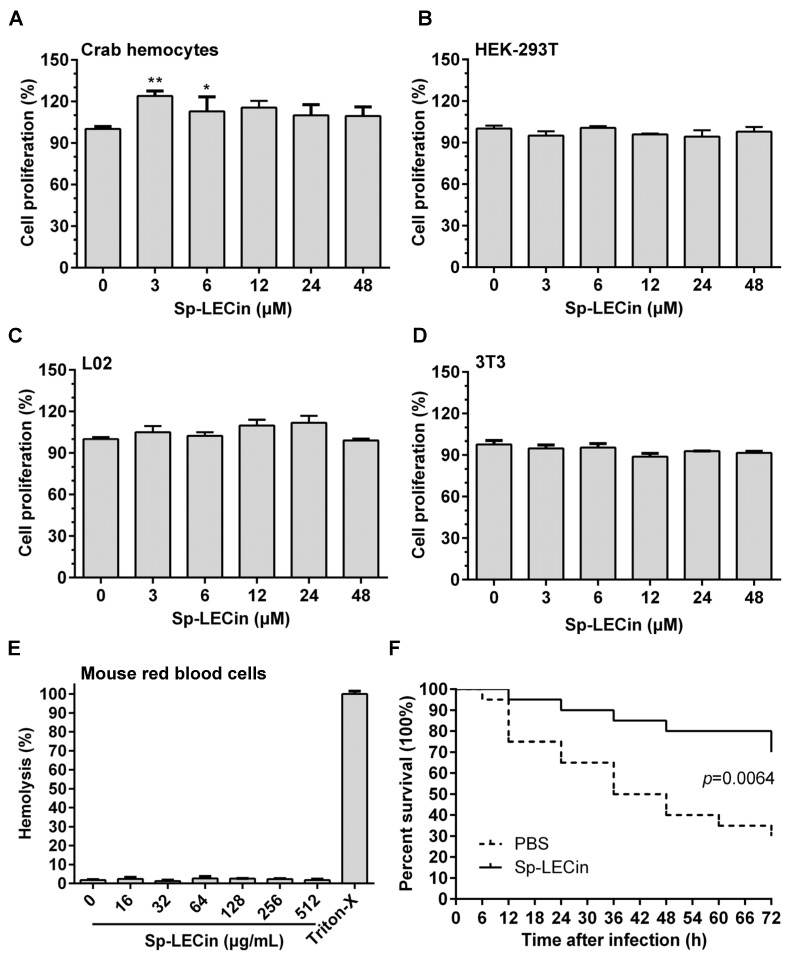
In vitro cytotoxicity and hemolytic activity and the in vivo effectiveness of Sp-LECin. The cytotoxicity of Sp-LECin on crab hemocytes (**A**), human embryonic kidney cells (HEK-293T) (**B**), human hepatic cells (L02) (**C**), and mouse embryonic fibroblast (3T3) (**D**), were determined by the MTS-PMS assay. (**E**) Hemolytic activity of Sp-LECin against mouse red blood cells. Significant difference between the control group and the Sp-LECin treatment group was indicated by asterisks as * *p* < 0.05 and ** *p* < 0.01. The bars indicated the mean ± standard error of mean (*n* = 3). (**F**) In vivo effectiveness of Sp-LECin in *P. aeruginosa*-challenged zebrafish (*n* = 20/group). The survival curve of each group was analyzed using the log-rank test.

**Table 1 ijms-24-00267-t001:** The sequence and key physicochemical parameters of Sp-LECin.

Physicochemical Parameters	Sp-LECin
Sequence	GCVFLLPAKPHNYKKVFLSKGV
Number of amino acids	22
Molecular weight	2.45 kDa
Total net charge	+4
Isoelectric point	9.87
Hydrophobicity	45%

**Table 2 ijms-24-00267-t002:** Antimicrobial activity of Sp-LECin.

Microorganism	CGMCC NO.	MIC (μM)	MBC/MFC (μM)
Gram-positive bacteria			
*Listeria monocytogenes*	1.10753	<3	3–6
*Enterococcus faecium*	1.131	3–6	12–24
*Enterococcus faecalis*	1.2135	6–12	12–24
*Staphylococcus aureus*	1.2465	24–48	24–48
*Staphylococcus epidermidis*	1.4260	12–24	24–48
*Bacillus subtilis*	1.3358	3–6	6–12
Gram-negative bacteria			
*Acinetobacter baumannii*	1.6769	6–12	6–12
*Pseudomonas aeruginosa*	1.2387	12–24	12–24
*Pseudomonas stutzeri*	1.1803	<3	6–12
*Pseudomonas fluorescens*	1.3202	6–12	12–24
*Escherichia coli*	1.2389	24–48	24–48
*Shigella fiexneri*	1.1868	3–6	6–12
*Filamentous fungi*			
*Fusarium oxysporum*	3.6785	12–24	12–24
*Fusarium solani*	3.5840	12–24	24–48
*Fusarium graminearum*	3.3490	24–48	24–48
*Aspergillus niger*	3.316	12–24	24–48
*Aspergillus fumigatus*	3.5835	12–24	48–96
*Aspergillus ochraceus*	3.5830	12–24	24–48

CGMCC NO., China General Microbiological Culture Collection Center Number. The values of MIC and MBC/MFC are expressed as the interval [a]–[b]. [a] is the highest concentration with visible microbial growth in the tested, and [b] is the lowest concentration with no visible microbial growth.

## Data Availability

The data presented in this study are available on request from the corresponding author.

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
