# Peer review of "A Novel Antimicrobial Peptide Sp-LECin with Broad-Spectrum Antimicrobial Activity and Anti-Pseudomonas aeruginosa Infection in Zebrafish"

_ijms, 2022, doi:10.3390/ijms24010267_

Round 1

Reviewer 1 Report

In this submission, the author has reported a chemically synthesized truncated peptide containing 22 amino acids derived from a C-type lectin homolog SpCTL6 of Scylla paramamosain was screened and reported to exhibit broad-spectrum antimicrobial activity. The author has reported that, unlike other antimicrobial peptides, Sp-LECin is nontoxic cytotoxicity and could greatly improve the survival of P. aeruginosa-infected zebrafish, by approximately 40% over the control group after 72 h of treatment. The manuscript has been well written and can be published after considering the following comments.

 Here are the comments which need to be addressed.

The selected peptide has four lysines as positively charged amino acids that are responsible for toxicity. However, why author is not seeing any toxicity? Can the author discuss any structure-property relation based on the observed results and chemical structure of the peptide? 

Reviewer 2 Report

The authors provide a complex study of a new antimicrobial peptide, a new potential antimicrobial. The manuscript is easy to read and to follow, it addresses interesting and significant issue of new antimicrobial agents development. However, there are some remarks concerning the text. 

1)Uncontrolled formation of ROS could be destructive not only for bacterial cells, but also for the host tissues. Which ROS-assosiated antibacterial mechanism do you propose for your AMP?

2) Lines 336-338. The rate of bactericidal effects could not prevent the bacterial resistance development. This could be seen from the example of the polymyxins antibiotics, where the resistance is also occurs despite the extremely high rates and effectiveness of action. Thus, I propose to reformulate or to remove this phrase. 

3) The nutrient broths composition as well as controls used should be described in details for each of experiments.

4) Why did you use crab hemocytes rather than human blood cells for cytotoxicity assay?

5) In my opinion the references throughout the text are excessive and cold be reduced.

Reviewer 3 Report

The manuscript entitled “A new antimicrobial peptide Sp-LECin with broad-spectrum 2 antimicrobial activity and anti-Pseudomonas aeruginosa infection in zebrafish” by Chen et al talks about the antimicrobial activity of the peptide and its potential role in anti-endotoxin activity as it binds to LPS. The work is novel but very preliminary. There is no in vivo study to truly asses the antimicrobial property of the peptide.

Major Comments Below:

1)      Section 2.5. This is not an LPS binding Experiment. Please consider performing a Limulus Amebocyte Lysate (LAL) assay. If that is not an option, try performing an ELISA experiment with mouse macrophages or THP-1 Cell line with LPS and the newly identified peptide. Or at least a Nitric Oxide (NO) estimation using a Griess Reagent. For protocols, Please refer to Selective phenylalanine to proline substitution for improved antimicrobial and anticancer activities of peptides designed on phenylalanine heptad repeat. Acta Biomater. 2017 Jul 15; 57:170-186.

2)       Consider doing the antibacterial assay in presence of human serum and report the MICs in serum. I see that the manuscript do not have the in vivo experiments. Hence, serum stability is the minimum requirement to reflect their true  anti-microbial potential in the living system.

Minor Comments:

1)      Change the heading  from “A new antimicrobial peptide” to “A Novel antimicrobial peptide”.

2)      Change hydrophobic rate to hydrophobicity.

3)      Line 145, Microorganisms instead of Microorganism

4)      Figure1. The Scanning Micrographs have different size bars. Please keep them uniform throughout.

5)      A. niger and F.oxysporum can have different size bars.

6)      Please perform the MTT on 3T3 cell lines.

7)      2.3 Instead of Killing Kinetics write Bactericidal Kinetics

Round 2

Reviewer 3 Report

I would like to congratulate the authors on doing a very meticulous job in revising the manuscript. 

I have one minor suggestion for them before it gets accepted in IJMS.

The sequence and the biophysical parameters of the peptide don't seem to be highlighted very well.

The reader will get to know the sequence after reading 411 lines of the manuscript.

I would request the authors to make a short table and mention the sequence of the peptide and other biophysical parameters they calculated such as net charge, percent hydrophobicity, etc. Please give the HPLC profile and the mass spectra of the peptide as well in the supplemental file, if possible. 
